# Enhancing the Therapeutic Potential of Human Umbilical Cord Mesenchymal Stem Cells for Osteoarthritis: The Role of Platelet-Rich Plasma and Extracellular Vesicles

**DOI:** 10.3390/ijms26083785

**Published:** 2025-04-17

**Authors:** Yu-Hsun Chang, Kun-Chi Wu, Dah-Ching Ding

**Affiliations:** 1Department of Pediatrics, Hualien Tzu Chi Hospital, Buddhist Tzu Chi Medical Foundation, Tzu Chi University, Hualien 970, Taiwan; cyh0515@gmail.com; 2Department of Orthopedics, Hualien Tzu Chi Hospital, Buddhist Tzu Chi Medical Foundation, Tzu Chi University, Hualien 970, Taiwan; drwukunchi@yahoo.com.tw; 3Department of Obstetrics and Gynecology, Hualien Tzu Chi Hospital, Buddhist Tzu Chi Medical Foundation, Tzu Chi University, Hualien 970, Taiwan; 4Institute of Medical Sciences, College of Medicine, Tzu Chi University, Hualien 970, Taiwan

**Keywords:** human umbilical cord mesenchymal stem cells, platelet-rich plasma, extracellular vesicles, osteoarthritis

## Abstract

Osteoarthritis (OA) is a chronic degenerative joint disease. Our previous study demonstrated that extracellular vesicles (EVs) secreted by human umbilical cord mesenchymal stem cells (HUCMSCs), which play a crucial role in regenerative medicine, have therapeutic effects on OA. Additionally, platelet-rich plasma (PRP) has been widely used in musculoskeletal diseases as it promotes wound healing, angiogenesis, and tissue remodeling; however, its efficacy as a stand-alone therapy remains controversial. Therefore, we investigated the therapeutic effects of combining stem cell-derived EVs with PRP in an OA model. HUCMSC-derived EVs treated with PRP were used as the experimental group, whereas HUCMSC-derived EVs cultured with serum-free (SF) or exosome-depleted fetal bovine serum (exo(-)FBS) and PRP served as controls. PRP-treated HUCMSCs maintained their surface antigen characteristics and potential to differentiate into adipocytes, osteoblasts, and chondrocytes. In the OA model, mice treated with HUCMSCs + 5% PRP-derived EVs showed significantly improved motor function compared to controls and were comparable to those treated with HUCMSCs +SF and +exo(-)FBS-derived EVs. Additionally, increased type II collagen and aggrecan and decreased IL-1β expression were observed in cartilage transplanted with various EVs. In conclusion, PRP enhances HUCMSC differentiation, whereas treatment with EVs improves OA outcomes, providing a promising strategy for future clinical applications.

## 1. Introduction

Osteoarthritis (OA) is a common chronic degenerative joint disease that primarily affects older individuals and is more common in women than in men [1]. Most cases of OA involve weight-bearing joints, such as the knees and hips, resulting in the destruction of articular cartilage and subchondral bone, along with the formation of osteophytes [2]. Although cartilage loss is central to the condition, OA affects the entire joint structure [3]. Knee OA presents symptoms such as pain, stiffness, limited joint mobility, and muscle weakness. Over time, it can lead to decreased physical activity, physical deconditioning, disrupted sleep, fatigue, depression, and increased disability [3]. Various factors, including occupational strain, participation in sports, musculoskeletal injuries, obesity, and sex, influence OA development [3,4].

Pharmacological treatments for symptomatic relief include acetaminophen, non-steroidal anti-inflammatory drugs, opioids, topical analgesics, and corticosteroid injections [5]. Additionally, intra-articular (IA) injections of hyaluronic acid (HA) and platelet-rich plasma (PRP) have been used to alleviate symptoms [6]; however, the efficacies of these treatments remain controversial. A recent systematic review indicated that IA HA injections could delay the need for total knee arthroplasty [7,8]. Additionally, another study suggested that intra-articular PRP injections might have a similar effect. Thus, the current data are insufficient to conclusively determine the impact of IA HA injections on delaying total knee arthroplasty [8].

Multipotent mesenchymal stem cells (MSCs) can differentiate into cells within a single germ layer [9]. Moreover, tissue-specific MSCs can repair damaged tissues and organs [10]. Human umbilical cord stem cells (HUCMSCs) are isolated from the umbilical cord and possess unique properties such as hypoimmunogenicity, absence of teratoma induction, and anticancer properties [11,12]. The ability to modulate immune responses makes HUCMSCs a crucial compatible stem cell source for transplantation therapy in allogeneic settings without immunorejection [13]. Consequently, MSCs have also been used in OA therapy. Additionally, PRP treatment has demonstrated better therapeutic effects in Kellgren–Lawrence (KL) grade I–II OA, while MSCs have shown superior outcomes in KL grade II–III OA [14]. Based on these clinical findings, researchers have investigated therapies that combine PRP with MSCs [15].

PRP is an autologous concentrate containing numerous growth factors (including transforming growth factor beta, platelet-derived growth factor, and vascular endothelial growth factor) that promote tissue regeneration, have anti-inflammatory effects, and stimulate chondrocytes to synthesize the extracellular matrix [16,17]. PRP is primarily used in OA to treat mild to moderate lesions [17]. However, its clinical efficacy remains debatable due to differences in preparation methods and formulation composition [17]. PRP used as an additive for culturing MSCs may affect their biological activity, including the profile of secreted extracellular vesicles (EVs), providing a potential basis for therapeutic effects in OA [17].

EVs are small vesicles containing abundant RNA molecules and proteins secreted by cells [18,19] that play crucial roles in physiological and pathological processes. EVs represent a cell-free therapy for OA that can reduce pain, repair cartilage, and reduce inflammation [20] and induce minimal immune responses [19,21]. Consequently, clinical investigations have explored combined therapies using PRP and MSCs for OA [22]. However, whether PRP enhances the therapeutic effects of HUCMSC EVs in OA is unknown.

Therefore, in this study, we investigated the effects of PRP on HUCMSCs and explored the therapeutic effects of EVs on PRP-treated HUCMSCs in OA.

## 2. Results

### 2.1. HUCMSCs Present Typical MSC Characteristics After PRP Treatment

Morphology and MSC surface markers were used to identify the characteristics of HUCMSCs. After treatment with 5% PRP for 24 h, HUCMSCs retained their typical fibroblast-like appearance (Figure 1A,B). The HUCMSCs were positive for CD73, CD90, CD105, and HLA-ABC but negative for CD34, CD45, and HLA-DR. Only a few cells expressed CD44 (Figure 1C,D). Based on these findings, we confirmed that HUCMSCs maintained their MSC characteristics after 24 h treatment with 5% PRP, as evidenced by their morphology and surface marker profiles [23].

### 2.2. PRP-Treated HUCMSCs Could Differentiate into Adipocytes, Osteoblasts, and Chondrocytes

HUCMSCs cultured in various conditions could differentiate into adipocytes, osteoblasts, and chondrocytes under specific differentiation media (Figure 2). Adipocyte differentiation was confirmed by Oil Red staining (Figure 2A) and increased expression of adipocyte-specific genes (*FABP4*, *PPAR-γ*) (Figure 2C). Osteoblast differentiation was confirmed by Alizarin Red staining (Figure 2A) and increased expression of osteoblast-specific genes (*ALPL*, *RUNX2*) (Figure 2C). Chondrocyte differentiation was confirmed by the pellet formation, type II collagen and aggrecan staining (Figure 2B), and increased expression of chondrocyte-specific genes (*ACAN, COL2A1*) (Figure 2C). These data suggest that PRP promotes the multilineage differentiation capacity of HUCMSCs.

### 2.3. Characterization of EVs Secreted by HUCMSCs

HUCMSCs were cultured in exosome-depleted fetal bovine serum (exo(-)FBS) or 5% PRP medium for 24 h. Subsequently, the cells were washed with PBS and cultured in a serum-free (SF) medium for 48 h, and the CM was collected for EV isolation and identification. The electron beam generated detailed images revealing the typical round, cup-shaped morphology of EVs, with diameters ranging from 30–150 nm (Figure 3A). The concentration of EVs secreted from HUCMSCs cultured with SF was 2.28 × 10^8^ particles/mL, and their mean size was 137.7 nm (Figure 3B, left panel). Similarly, the concentration of EVs secreted from HUCMSCs cultured with 5% PRP medium was 2.59 × 10^8^ particles/mL, and their mean size was 89 nm (Figure 3B, middle panel).The concentration of EVs secreted from HUCMSCs cultured with (exo(-)FBS) was 1.59 × 10^10^ particles/mL, and their mean size was 127.5 nm (Figure 3B, right panel). Western blot analysis revealed higher expression of CD9, CD63, CD81, and GM130 in EVs secreted from HUCMSCs cultured in 5% PRP medium than those cultured in (exo(-)FBS) (Figure 3C).

### 2.4. EVs Secreted from PRP-Treated HUCMSCs Could Attenuate the OA

A collagenase-induced OA model was established using 10-week-old C57BL/6 mice undergoing rotarod training to explore the therapeutic potential of EVs secreted from PRP-treated HUCMSCs (OA_112WJ_5%PRP) in OA. On day 0, 12 U of collagenase VII was injected intraarticularly into the patellar ligament. On day 7, the mice in the study group were treated with EVs extracted from 5% PRP-cultured HUCMSCs. The positive control group received EVs from (exo(-)FBS)-cultured HUCMSCs (OA_EXO) or EVs extracted from PRP (OA_5%PRP). The normal and negative control groups were treated with saline.

The walking capacity of mice following knee injury and treatment using the rotarod test was evaluated. Mice in each group underwent a rotarod performance test on days 0, 7, 14, 21, and 28. Notably, mice treated with EVs extracted from 5% PRP-cultured HUCMSCs exhibited significantly greater improvement after day 14 than the negative control group. The recovery was similar to that of the normal and positive control groups (Figure 4).

### 2.5. Improvement in the Histology of Cartilage Recovery After EV Transplant

Figure 5 displays histological images of joint tissues stained with H&E and Safranin O, demonstrating enhanced cartilage integrity and proteoglycan retention in the HUCMSC-SF-e and +5%PRP-e groups, but not in the PRP-e-treated groups compared to the OA group. Among these, the +5%PRP-e group exhibited the most pronounced improvement (presence of proteoglycans in the cartilage matrix). Lower Safranin O staining (reduced presence of proteoglycans in the cartilage matrix) was noted in the 5% PRP-e and OA groups.

### 2.6. Improvement in International Cartilage Repair Society Histological Score After EV Transplant

Figure 6 illustrates the International Cartilage Repair Society (ICRS) histological scores across different treatment groups, comparing the normal, OA, MSC + 5%PRP-e, 5%PRP-e, and HUCMSC-e groups. The OA group exhibits the lowest ICRS score, indicating severe cartilage degradation. In contrast, the MSC + 5%PRP-e group shows a significant improvement (*p* < 0.001), reaching scores comparable to the normal group. Similarly, the HUCMSC-e group also demonstrates a statistically significant increase (*p* < 0.01), though to a lesser extent. These results suggest that MSC + 5%PRP-e exosomes provide the most effective cartilage repair, followed by HUCMSC-e, while 5%PRP-e alone has a limited effect in restoring cartilage integrity.

### 2.7. Transplantation of EVs Increased Type II Collagen and Aggrecan Expressions

Figure 7 shows the immunohistochemistry (IHC) staining results of type II collagen and aggrecan expression in the various treatment groups. IHC staining for type II collagen revealed reduced expression in the OA group compared to that in the normal group, indicating cartilage degradation (Figure 7A,B). Conversely, treatment with +5% PRP-e or HUCMSC-e restored type II collagen expression, suggesting a protective effect (Figure 7A,B). Similarly, the results showed decreased aggrecan expression in the OA group and increased expression in the treatment groups, indicating enhanced cartilage repair (Figure 7C). These findings suggest that 5%PRP-e, +5%PRP-e, and HUCMSC-e promote cartilage regeneration by enhancing the expression of type II collagen and aggrecan.

### 2.8. EV Transplant Decreased IL-1beta Expressions

Figure 8 illustrates the effects of different treatments on OA using IHC for IL-1β and MMP-13. Increased IL-1β staining indicates increased inflammation in the OA group, while treatment with +5% PRP-e and HUCMSC-e shows reduced IL-1β expression (Figure 8A,B). However, cartilages stained with MMP13 did not differ significantly among the treatments (Figure 8C,D). Thus, PRP- and HUCMSC-derived EVs help suppress inflammation and restore cartilage matrix proteins but may not directly inhibit matrix degradation.

## 3. Discussion

In this study, we investigate PRP’s effects on HUCMSCs and EVs’ therapeutic effects in OA. We found that HUCMSCs exhibited typical MSC characteristics and differentiation capability after PRP treatment. Compared with HUCMSCs cultured with exo(-)FBS, those cultured in 5% PRP secreted fewer EVs. In a collagenase-induced OA model, mice treated with EVs extracted from 5% PRP-cultured HUCMSCs showed significantly better exercise function during the rotarod performance test than those without treatments. The cartilage recovery was similar to that of the normal and positive control groups. The IHC of the cartilage also demonstrated increased type II collagen and aggrecan expression and reduced IL-1β expression after EV treatment.

Figure 7 demonstrates that IHC staining for type II collagen and aggrecan—key markers of cartilage matrix integrity—was markedly reduced in the OA group, confirming matrix degradation typical of osteoarthritis [24]. Notably, treatment with +5% PRP-e or HUCMSC-e significantly restored the expression of both type II collagen and aggrecan, suggesting a regenerative effect that supports cartilage repair. These findings imply that +5%PRP-e and HUCMSC-e can mitigate cartilage breakdown and actively promote matrix synthesis. Figure 8 further complements these observations by showing reduced IL-1β expression in the treatment groups, highlighting an anti-inflammatory effect [25]. However, the lack of significant differences in MMP-13 staining across groups suggests that while +5%PRP-e and HUCMSC-e can reduce inflammatory cytokines and enhance matrix protein expression, they may have limited efficacy in directly suppressing matrix-degrading enzymes [26]. These results collectively underscore the dual function of these treatments in modulating inflammation and supporting matrix regeneration.

Typically, most umbilical cords are discarded after delivery, except in the case of personal banking [27,28]. Owing to the abundance of HUCMSCs, they can meet cell transplantation needs throughout an individual’s life [12]. Furthermore, HUCMSCs originating from epiblasts of the human embryo and Wharton’s jelly in the human umbilical cord exhibit multipotency and the ability to differentiate into various human cell types, including neural-like cells, osteocytes, and adipocytes, both in vitro and in vivo [29]. Similarly, exosome-depleted FBS-cultured HUCMSCs exhibit classic MSC characteristics, such as enhanced chondrogenic differentiation and improved cartilage repair in an OA mouse model, suggesting their potential for future clinical applications in osteoarthritis treatment [30]. Moreover, fetal MSCs exhibit better differentiation abilities than adult MSCs [31].

PRP contains numerous growth factors, cytokines, lysosomes, and adhesion proteins that initiate healing in injured tissues [32]. These growth factors stimulate the proliferation and differentiation of stem cells in injury models [33]. A meta-analysis of randomized controlled trials investigated the efficacy and safety of MSCs combined with PRP in treating knee OA [22]. The results showed that this combined therapy improved pain and joint function more effectively without significantly increasing the adverse effects. These studies highlight the potential of PRP for enhancing MSC function and promoting tissue regeneration. Notably, PRP is autologous and eliminates the risk of immune rejection.

EVs derived from HUCMSCs also show potential for OA treatment as they exhibit anti-inflammatory and immunomodulatory effects by promoting M2 macrophage polarization and modulating the PI3K-Akt signaling pathway [34]. Additionally, they maintain chondrocyte homeostasis and alleviate OA pathology via the miR-223/NLRP3/pyroptosis axis [35]. Accordingly, a previous study reported that engineering EVs with miR-7704 improved cartilage repair and reduced MMP13 expression in OA models [36]. Thus, HUCMSC-derived EVs offer several advantages over whole-cell therapies, including lower immunogenicity and tumorigenicity [37], suggesting their potential as a novel therapeutic approach for OA, with mechanisms involving inflammation reduction, cartilage protection, and targeted delivery of beneficial miRNAs to the affected joints.

Platelet-derived EVs demonstrate superior regenerative potential compared with mesenchymal stem cell-derived EVs in both ex vivo and in vivo OA models [38]. EVs from autologous blood products, such as PRP and hyperacute serum, can enhance chondrogenic gene expression and reduce inflammation in OA chondrocytes [39]. MSC- and PRP-derived EVs show therapeutic potential by suppressing inflammation and reducing chondrocyte apoptosis [40]. EVs can be used for treatment, with cell pre-treatment strategies and EV tissue engineering playing increasingly important roles in OA therapy [41].

The biological effects of EVs can be influenced by both particle size and concentration. Exosomes are extracellular vesicles ranging from 30 to 150 nm in size that play a crucial role in intercellular communication [42,43]. Their biological effects are influenced by particle size, with smaller exosomes showing faster uptake by target cells and increased cellular motility [44]. Exosomes contain various biomolecules, including proteins, nucleic acids, and lipids, which can affect recipient cell function [42,43]. Their non-cytotoxic nature and ability to carry diverse cargo make them promising candidates for drug delivery and personalized medicine [45]. Studies have shown that EV concentration impacts gene expression and protein levels in target cells, with different effects observed at low versus high concentrations [46]. Sample concentration is critical for obtaining reproducible data in nanoparticle tracking analysis of EVs, with increased variability at higher concentrations due to particle interactions [47]. These findings highlight the importance of considering EV concentration and size distribution in both research and potential clinical applications.

Recent studies have highlighted the need for long-term follow-up of MSC therapy for OA. A seven-year study on knee OA patients treated with adipose-derived MSCs showed significant clinical improvements for up to 60 months, with sustained cartilage structural benefits at 84 months [48]. However, an ovine model study found that while intra-articular MSC injections decelerated OA progression, no significant differences were observed between treated and control groups after 12 weeks [49]. The development of disease-modifying OA drugs remains a priority [50]. Safety concerns have been addressed in a 30-month follow-up study of 91 patients treated with autologous adipose-derived stem cells and PRP, which reported no neoplastic complications and only minor, self-limiting side effects [51]. These findings underscore the importance of continued research into the long-term efficacy and safety of stem cell or EV treatments for OA.

Recent studies have explored various OA models to understand the disease’s pathogenesis better and evaluate potential treatments. The collagenase-induced OA model in rats and mice is used to study the progression of OA and potential therapies. This model mimics slow-progressing human OA, characterized by cartilage degeneration and synovial inflammation [52,53]. The destabilization of the medial meniscus (DMM) model has been widely used in mice, inducing structural changes in articular cartilage, subchondral bone, and menisci [54]. This model has been successfully applied in humanized mice, offering a promising tool for testing immune-interacting therapies [55]. While animal models remain valuable, engineered tissue models are emerging as potential alternatives for studying OA pathogenesis and evaluating treatments [56]. These diverse approaches contribute to a more comprehensive understanding of OA and may improve therapeutic strategies.

### Strengths and Limitations

A strength of this study is that PRP enhanced the differentiation potential of HUCMSCs, while their EVs contributed to cartilage regeneration in an OA model. This study confirmed that PRP-treated HUCMSCs retain their surface antigen characteristics and trilineage differentiation ability, highlighting the feasibility of using PRP as a culture supplement to enhance stem cell therapy. Additionally, the therapeutic effects of EVs on OA, including improved motor function, increased type II collagen and aggrecan expression, and reduced IL-1β levels, support the potential of EV-based regenerative treatments. This study also provides comparative analyses under different culture conditions, strengthening the validity of PRP’s role of PRP in HUCMSC differentiation and its subsequent EV-mediated effects.

However, this study has some limitations. Although PRP-enhanced EVs showed therapeutic benefits, their effects were comparable to those of EVs from other culture conditions (SF- and exo(-)FBS), suggesting that PRP treatment may not provide a significant advantage. Therefore, future studies should include a long-term evaluation of cartilage integrity and functional recovery to assess the durability of PRP-enhanced EV therapy for OA.

## 4. Materials and Methods

### 4.1. Ethics

HUCMSCs were selected for this study based on our extensive experience with their derivation and culture. The experimental protocol was approved by the Research Ethics Committee of Buddhist Tzu Chi General Hospital (IRB 111-231-B), and all participants provided written informed consent.

### 4.2. HUCMSC Culture and Identification of the MSC Characteristics

HUCMSCs were derived according to the previously published protocol [57] using HUCMSC lines obtained from a previous study. In brief, the HUCMSCs were cultured in Dulbecco’s Modified Eagle Medium (DMEM)—low glucose supplemented with fetal bovine serum (FBS)/10% exosome depleted (exosome(-))FBS/serum-free medium with 5% PRP and 1% Penicillin-Streptomycin antibiotics at 37 °C in a 95% air and 5% CO_2_ humidified atmosphere. The HUCMSCs were characterized using flow cytometry and differentiation assays to confirm their MSC characteristics.

### 4.3. Platelet-Rich Plasma Preparation

Whole blood from Sprague Dawley rats was procured from BioLASCO Taiwan Co., Ltd. (Taipei, Taiwan). Blood samples were collected in a sterile controlled environment. Subsequently, blood was transferred into tubes containing 3.8% sodium citrate. The tubes were then centrifugated at 2300× rpm for 15 min. After centrifugation, the supernatant (PRP) was carefully collected and stored at −80 °C for future use. This protocol ensured the isolation of PRP containing essential growth factors and bioactive molecules for potential therapeutic applications [58].

### 4.4. Flow Cytometry

Flow cytometry was used to evaluate the surface marker expression of HUCMSCs at passages 3–4. HUCMSCs were isolated by treatment with phosphate-buffered saline (PBS) containing Accutase (Millipore, Billerica, MA, USA) and washed with PBS containing 2% bovine serum albumin and 0.1% sodium azide (Sigma-Aldrich, St. Louis, MO, USA). Thereafter, the cells were incubated with primary antibodies labeled with either phycoerythrin or fluorescein isothiocyanate targeting surface markers such as CD34, CD44, CD45, CD73, CD90, CD105, HLA-ABC, and HLA-DR (BD Biosciences, Franklin Lakes, NJ, USA). Analysis was performed using a Becton Dickinson flow cytometer (Becton Dickinson, San Jose, CA, USA).

Subsequently, the HUCMSCs were divided into two groups:

Control Group: HUCMSCs were cultured with SF.

Study Group: HUCMSCs were cultured in a medium containing 5% PRP for 24 h.

These experimental groups allowed us to investigate the effects of PRP on HUCMSC characteristics.

### 4.5. HUCMSC Trilineage Differentiation

The protocols in Section 2.6, Section 2.7 and Section 2.8 were employed to differentiate HUCMSCs in the control and study groups into adipocytes, osteocytes, and chondrocytes. The HUCMSCs were divided into five groups:

Control Group 1: HUCMSCs cultured in a medium containing 10% FBS for 24 h.

Control Group 2: HUCMSCs cultured in SF medium for 24 h.

Control Group 3: HUCMSCs cultured in a medium containing 10% exo(-)FBS for 24 h.

Study Group: HUCMSCs were cultured in a medium containing 5% PRP for 24 h.

These experimental groups allowed us to investigate the effects of PRP on HUCMSC behavior.

### 4.6. Adipocyte Differentiation

The adipocyte differentiation medium consisted of DMEM—high glucose supplemented with 10% FBS, 0.5 mM IBMX (3-isobutyl-1-methylxanthine), 1 μM dexamethasone, 60 μM indomethacin, and 5 μg/mL insulin (all obtained from Sigma). HUCMSCs were plated at a density of 5 × 10^4^ cells/well in a 12-well plate and cultured in an adipogenic medium for 14 days, with medium replacement twice weekly. After 14 days, cells from all groups were stained with Oil Red O (Sigma). Additionally, cell samples were harvested for *FABP4* and *PPARγ* gene analysis using quantitative real-time polymerase chain reaction (qRT-PCR).

### 4.7. Osteocyte Differentiation

The osteocyte differentiation medium comprised DMEM—high glucose supplemented with 10% FBS, 0.1 μM dexamethasone, 50 μM LAS2P, and 10 μM BGP. HUCMSCs were seeded at a density of 1 × 104 cells per well in a 12-well plate and cultured in an osteogenic medium for 14 days, with the medium replaced twice per week. After 14 days, cells from all groups were stained with Alizarin Red. Cell samples were harvested for *RUNX2* and *APAL* gene analysis using qRT-PCR.

### 4.8. Chondrocyte Differentiation

The chondrocyte differentiation medium comprised DMEM—high glucose supplemented with 10% FBS, 0.1 μM dexamethasone, 50 μM LAS2P, 40 μM L-proline, 100 μg/mL sodium pyruvate, 10 ng/mL TGF-β3, and Insulin–Transferrin–Selenium (ITS) (all obtained from Sigma). Chondrogenesis was induced using the pellet culture method, where 1 × 10^6^ HUCMSCs were seeded in a 15 mL conical tube (BD Biosciences, Franklin Lakes, NJ, USA) containing 2 mL of chondrogenic medium for 21 days, with medium replacement every 2 days. After 21 days, the resulting pellets were photographed and fixed in 4% paraformaldehyde at 4 °C for 24 h. The pellets were washed with PBS and transferred to a 70% ethanol solution. Histological analyses were used to characterize the differentiated chondrocytes, including hematoxylin and eosin (H&E) and Safranin O staining. Additionally, cell samples were harvested for *ACAN* and *COL2A1* gene analysis using qRT-PCR.

These differentiation protocols allowed us to explore the behavior and characteristics of HUCMSCs under specific conditions.

### 4.9. qRT-PCR

RNA for all qRT-PCR analyses was quantified using the PureLink™ RNA Mini Kit (Invitrogen, Waltham, MA, USA). Briefly, 500 ng of RNA was treated using the GScript First-Strand Synthesis Kit grade (GeneDirex, Taichung, Taiwan). The FastStart Universal SYBR Green Master (ROX, Basel, Switzerland) gene expression assay and the ABI Step One Plus system (Applied Biosystems, Waltham, MA, USA) were used for qRT-PCR analysis. The housekeeping gene GAPDH served as an internal control. The primer sequences and product sizes for adipogenic (*PPARγ* and *FABP4*), osteogenic (*ALPL* and *RUNX2*), and chondrogenic genes (*ACAN* and *COL2A1*) in the qRT-PCR analysis are listed in Table 1. These protocols allowed us to investigate the gene expression patterns and cellular differentiation.

### 4.10. EV Isolation and Identification

HUCMSCs (5 × 10^6^) were seeded into a 5-layer flask and cultured in exosome-depleted FBS medium or 5% PRP medium for 24 h. Subsequently, the cells were washed with PBS and cultured in a serum-free medium for 48 h. The conditioned medium (CM) was collected and centrifuged at 1500× rpm for 5 min to remove the cells and cell debris. The supernatant was centrifuged again at 2000× *g* for 10 min and collected.

Thereafter, up to 15 mL of the sample, filtered using a 0.22 filter, was added to a centrifuge tube (Amicon^®^ Ultra-15 Centrifugal Filter Devices, 3 K, Millipore, Burlingon, MA, USA). The capped filter device was placed in the centrifuge rotor to ensure a counterbalance with a similar device. The device was spun at 3270× *g* for 40 min using a swinging bucket rotor. The concentrated solute was recovered, a pipettor was inserted at the bottom of the filter device, and the sample was withdrawn using a side-to-side sweeping motion to ensure complete recovery. The ultrafiltrate was then stored in a centrifuge tube.

Subsequently, the supernatant was transferred to a sterile vessel, and an appropriate volume of ExoQuick-TC (EXOTC50A-1, SBI, Palo Alto, CA, USA) was added to the biofluid. This well was mixed by inversion (Bio-fluid: ExoQuick = 5:1). Additionally, approximately 100 mL of the CM was concentrated and recovered to approximately 5 mL and refrigerated overnight at 4 °C. The ExoQuick-TC/biofluid mixture was centrifuged at 2200× *g* for 30 min. The supernatant was aspirated, and the residual ExoQuick-TC solution was centrifuged at 1500× *g* for 5 min. Finally, all fluid traces were removed by aspiration and the EV pellet was resuspended in 100–500 µL using 1× Exosome storage buffer (EV-GuardTM EV Storage Buffer EXSBA-1, System Biosciences, Palo Alto, CA, USA).

### 4.11. Characterization of EVs Using Nanoparticle Tracking Analysis

The suspensions containing EVs were analyzed using a Nano-Sight NS300 instrument (Malvern, Worcestershire, UK). Samples were diluted with 0.2 μm filtered 1× Normal Saline to achieve an optimal range of 20–150 particles/frame (typically diluted at 100× or 1000×). Three videos of 60 s each were recorded per sample, and the data were processed using NanoSight NTA Software 3.2 (Malvern). Post-acquisition settings were consistently maintained across all samples. The obtained data included the particle concentration (particles/mL), average size (nm), and particle size distribution.

### 4.12. Western Blot Analysis

Western blotting was performed to evaluate the expression of CD9, CD63, CD81, and GM130. GAPDH was used as an internal control. EVs were lysed using a protein lysis buffer (Sigma-Aldrich) to extract proteins, which were then separated using 10% sodium dodecyl sulfate-polyacrylamide gel electrophoresis (Sigma-Aldrich). Membranes were incubated overnight at 4 °C with primary antibodies against CD9, CD63, and CD81 (all from Sigma-Aldrich) at a 1:2000 dilution. Subsequently, the membranes were treated with a horseradish peroxidase-conjugated secondary antibody (Sigma-Aldrich) at a 1:5000 dilution. Horseradish peroxidase signals were detected using an electrochemiluminescence kit (Promega, Madison, WI, USA).

### 4.13. Transmission Electron Microscopy

Transmission electron microscopy (TEM) was used to visualize EVs, providing high-resolution images of their ultrastructure after isolation and purification. The vesicles were fixed in 2% glutaraldehyde (Sigma-Aldrich, St. Louis, MO, USA) and prepared in a phosphate buffer (pH 7.4; Sigma-Aldrich, St. Louis, MO, USA) to preserve their structural integrity. A 3 µL drop of the fixed EV suspension was applied to a carbon-coated TEM grid (697745, Sigma-Aldrich, St. Louis, MO, USA), followed by washing and staining with 1% uranyl acetate (21447-25, Polysciences, Warrington, PA, USA) as a contrasting agent. After air-drying, the grid was examined by TEM (HITACHI H-7500, Tokyo, Japan). The morphology, size, and structural characteristics of EVs were documented.

### 4.14. Collagenase-Induced Osteoarthritis Model

A collagenase-induced OA model was established as described previously [59]. Knee joints of 10-week-old C57BL/6 mice that had passed the rotarod training were injected with 12 U of collagenase VII (Clostridium histolyticum; Sigma-Aldrich, St. Louis, MO, USA) in 8 μL saline once intraarticularly through the patellar ligament on day 0. The mice were then divided into the following groups:

Normal group (3 mice): mice treated with 8 μL saline (Saline_ctrl).

Negative control group (3 mice): collagenase-induced OA mice treated with 8 μL saline (OA).

Positive control group 1 (6 mice): collagenase-induced OA mice treated with EVs extracted from 10% exo(-)FBS-cultured HUCMSCs (OA_EXO).

Positive control group 2 (6 mice): collagenase-induced OA mice treated with PRP EVs (OA_5%PRP)

Study group (6 mice): Collagenase-induced OA mice treated with EVs extracted from 5% PRP-cultured HUCMSCs (OA_112WJ_5%PRP).

### 4.15. Extracellular Vesicle Injection

Seven days after collagenase injection, mice in the three treatment groups received EVs at a concentration of 1 × 10^7^ particles/mL, delivered in 50 μL of PBS. The injections were performed under anesthesia using ketamine (50 mg/kg) and xylazine (15 mg/kg). EVs were administered at sites located beneath the infrapatellar ligament in both hind knees. Following the injections, mice were allowed free movement and ad libitum access to food and water in their cages.

### 4.16. Function Assessments

Functional assessments were performed on day 0 before saline/collagenase VII treatment, on day 7 before cell transplantation, and on days 14, 21, and 28 after transplantation. The mice were subjected to a rotarod performance test on the days the functional assessments were conducted.

### 4.17. Macroscopic Examination

The joint surfaces were grossly examined after the mice were euthanized using CO_2_ and decapitation on day 28. The distal femur and proximal tibial surfaces were exposed and examined macroscopically.

### 4.18. Histological Evaluation

The distal femur and the proximal tibial plateau were removed. After fixation with 10% buffered formalin (Sigma-Aldrich) for 48 h, the specimens were decalcified with 10% EDTA (Gibco, Grand Island, NY, USA) for 2 weeks and cut into four pieces. All pieces were embedded in paraffin. Serial sagittal sections were prepared and stained with H&E (Sigma-Aldrich) and Safranin O (Sigma-Aldrich). The histological changes were observed under a microscope. The sections were examined and evaluated in a blinded manner using a semi-quantitative grading and staging system which included six histological grades and four histological stages. The total score (grade multiplied by stage) ranged from 1 point (normal articular cartilage) to 24 points (no repair). A score of 0 indicated cartilage equivalent to normal tissue, and 3 indicated marked damage. Sections stained with H&E were scored based on the general morphology as 0 (normal hyaline cartilage), 1 (slightly reduced, <1/4), 2 (markedly reduced, 1/4–3/4), or 3 (no metachromatic staining, >3/4 reduction). Safranin O-stained slides were scored for glycosaminoglycan content as follows: 0 (normal), 1 (slightly reduced, 1/4), 2 (markedly reduced, 1/4–3/4), or 3 (no metachromatic staining, >3/4 reduction). A total score was determined by summing the scores of both staining methods to generate a total score where 0 represented normal, and 6 represented marked damage [60].

### 4.19. Immunohistochemical Staining

Immunohistochemical staining was performed on two distal femora per group. The femora were harvested and rapidly embedded in paraffin (Sigma). The staining targeted type II collagen, aggrecan, IL1-β, and MMP13 to assess the inflammatory status of joints in each group. The paraffin-embedded sections were initially deparaffinized with xylene and rehydrated through a series of graded ethanol solutions. Antigen retrieval was performed by heating the sections in citrate buffer (pH 6.0) at 95 °C for 20 min. After cooling to room temperature, endogenous peroxidase activity was quenched using 3% hydrogen peroxide, followed by blocking with 5% bovine serum albumin (BSA) to minimize nonspecific binding. The sections were then incubated overnight at 4 °C with primary antibodies against type II collagen, aggrecan, IL-1β, and MMP-13 (1:100, GeneTex, Irvine, CA, USA). Following PBS washes, the sections were treated with an HRP-conjugated secondary antibody (GeneTex) for 1 h at room temperature. Signal detection was achieved using a DAB substrate kit, and nuclei were counterstained with hematoxylin. Stained sections were examined under a light microscope (Nikon TE2000-U, Tokyo, Japan) for qualitative and quantitative analysis, with the percentage of positively stained cells determined by assessing five randomly selected fields.

### 4.20. Statistical Analysis

All data are expressed as medians, ranges, or means ± standard deviations. Statistical comparisons of the histopathological grades among the groups were conducted using non-parametric tests, such as the Mann–Whitney U test or analysis of variance (ANOVA) with post hoc analysis. Differences were considered significant when *p* < 0.05. All statistical analyses were performed using SPSS version 25 (IBM Corp., Armonk, NY, USA).

## 5. Conclusions

EVs derived from HUCMSCs with or without 5% PRP treatment exhibited therapeutic potential for treating OA by preserving cartilage integrity, promoting chondrocyte proliferation, and reducing joint inflammation. These findings highlight the regenerative capabilities of EV therapy and its potential role in the management of OA. Further research is warranted to optimize the EV delivery methods and long-term outcomes in clinical settings.

## Figures and Tables

**Figure 1 ijms-26-03785-f001:**
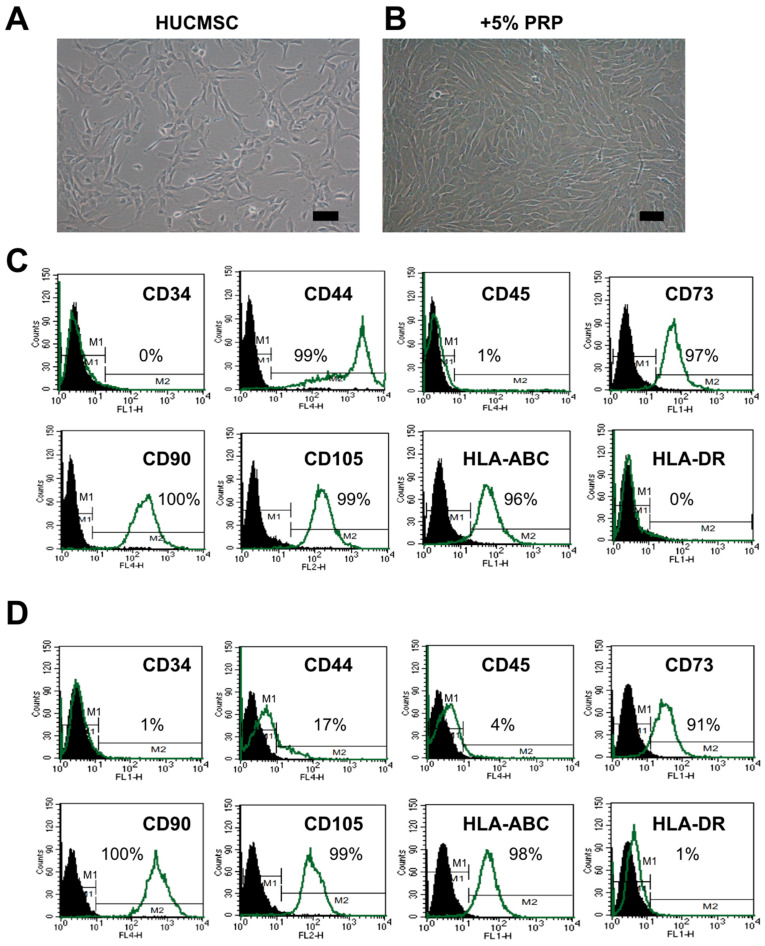
Characteristics of human umbilical cord mesenchymal stem cells (HUCMSCs) with/without platelet-rich plasma (PRP) treatment. (**A**) Morphology of HUCMSCs. Scale bar = 100 μm. (**B**) Morphology of HUCMSCs after treatment with 5% PRP for 24 h. (**C**) Flow cytometry of HUCMSCs. (**D**) Flow cytometry of HUCMSCs after treatment with 5% PRP for 24 h.

**Figure 2 ijms-26-03785-f002:**
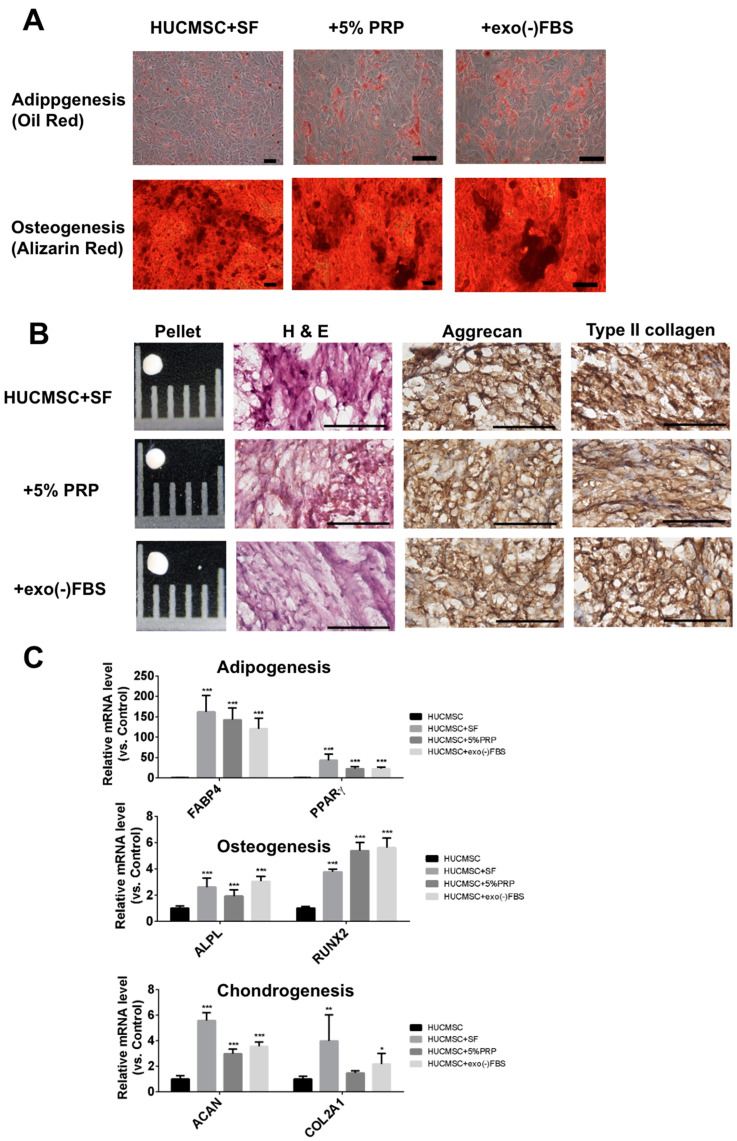
Trilineage differentiation of HUCMSCs in various culture media. (**A**) Representative images show adipogenic and osteogenic differentiation of HUCMSCs cultured in serum-free (SF) medium, 5% platelet-rich plasma (PRP), or exosome-depleted fetal bovine serum (exo(-)FBS), as assessed by Oil Red O staining for lipid accumulation and Alizarin Red staining for mineral deposition. Scale bar = 100 μm. (**B**) Chondrogenic differentiation was evaluated using pellet culture, with hematoxylin and eosin (H&E) staining confirming tissue formation and immunohistochemical staining for aggrecan and type II collagen, indicating extracellular matrix production. Scale bar = 100 μm. (**C**) Quantitative real-time PCR (qRT-PCR) analysis of adipogenesis (FABP4, PPAR-γ), osteogenesis (ALPL, RUNX2), and chondrogenesis (ACAN, COL2A1) markers. * *p* < 0.05, ** *p* < 0.01, *** *p* < 0.001.

**Figure 3 ijms-26-03785-f003:**
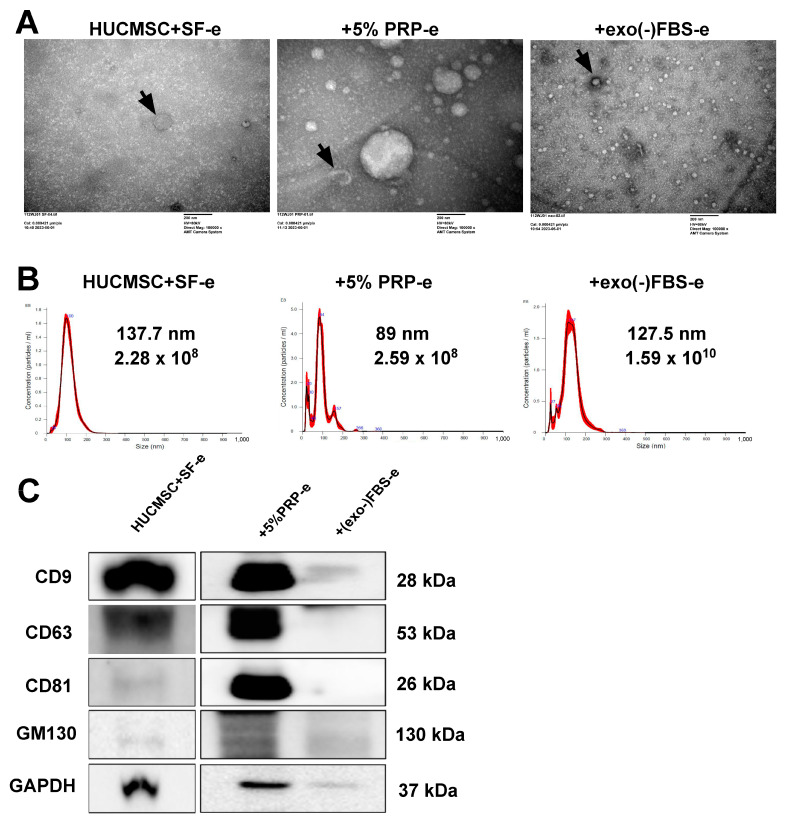
Identification of EVs secreted from HUCMSCs cultured with serum-free (SF), exosome-depleted FBS [(exo(-)FBS)] medium or 5% PRP medium. (**A**) EVs secreted by HUCMSCs were cultured in various media under an electron microscope (arrow). Scale bar = 200 nm. (**B**) Mean size and particle numbers of EVs derived from HUCMSCs cultured with multiple media. The **dark line** shows the actual measured data points from the instrument. The **red line** overlays a statistical fit or smoothing of that data to highlight the main particle size peak. (**C**) The Western blot shows positive CD9, CD63, CD81, and GM130 expression in EVs secreted by HUCMSCs cultured in various media.

**Figure 4 ijms-26-03785-f004:**
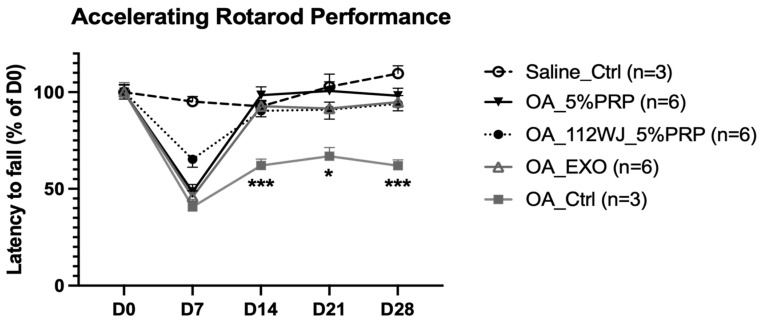
Rotarod performance test. The mice treated with EVs extracted from 5% PRP-cultured HUCMSCs (OA_112WJ_5%PRP) exhibited significantly better improvements after day 14 than the negative control group (OA_ctrl). Their recovery was similar to that of the normal (Saline_ctrl) and positive control groups (OA_EXO [+exo(-)FBS], OA_5%PRP). * *p* < 0.05, *** *p* < 0.001.

**Figure 5 ijms-26-03785-f005:**
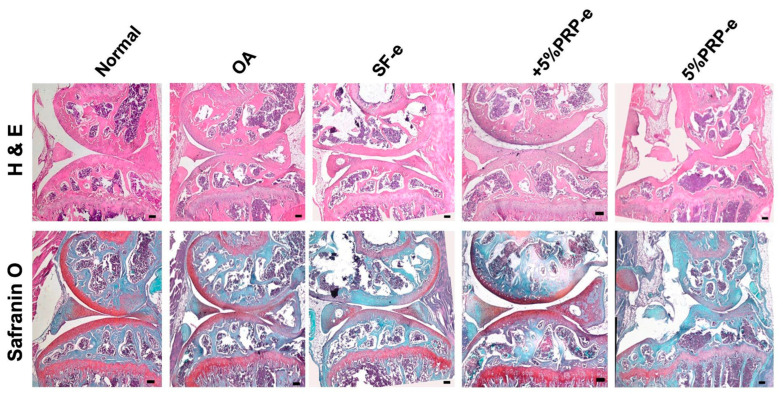
Histology of mouse OA model after 28 days of experiments. Representative images are illustrated. The upper panel shows H&E staining and the lower panel shows Safranin O staining. The mice were divided into normal knee joint and OA joint groups; OA mice underwent HUCMSC-EV (SF-e), HUCMSC + 5% PRP-EV, and 5% PRP-EV transplantation. Scale bar = 100 μm.

**Figure 6 ijms-26-03785-f006:**
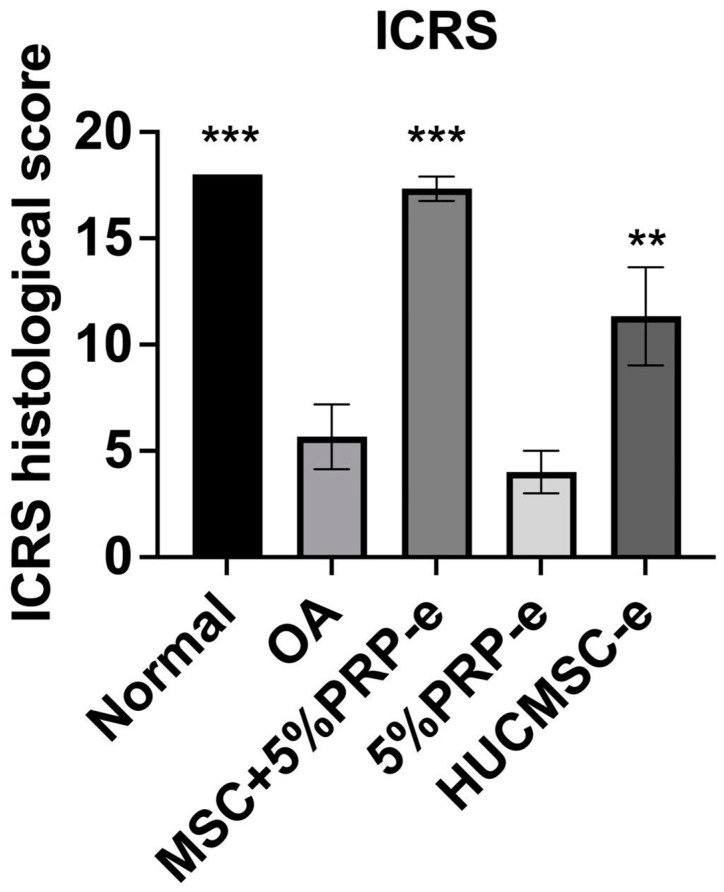
The International Cartilage Repair Society (ICRS) score of the mouse osteoarthritis (OA) model after 28 days of experiments. The ICRS score improved in the human umbilical cord mesenchymal stem cells (HUCMSC) + 5%PRP-EVs and HUCMSC-EVs treatment groups compared to the OA group. ** *p* < 0.01, *** *p* < 0.001.

**Figure 7 ijms-26-03785-f007:**
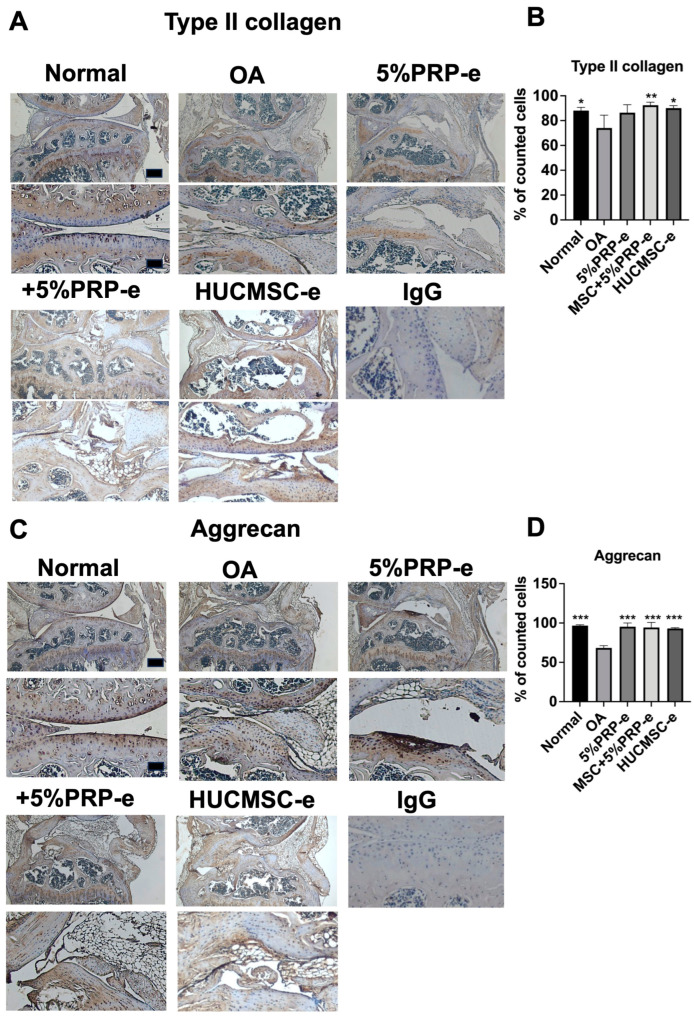
Immunohistochemistry of type II collagen and aggrecan in the mouse osteoarthritis model after 28 days of experiments. (**A**) A representative image of type II collagen is illustrated from the normal knee, osteoarthritis (OA), and 5% PRP-e, MSC + 5% PRP-e, and MSC-e treated groups (n = 3 in each group). Scale bar = 100 μm. The lower panel is the magnification of the upper panel. Scale bar = 250 μm. (**B**) Quantitative analysis of the percentage of positively stained cells. (**C**) A representative image of aggrecan is illustrated from the normal knee, OA, 5% PRP-e, MSC + 5% PRP-e, and MSC-e groups (n = 3 in each group). The lower panel is the magnification of the upper panel. (**D**) Quantitative analysis of the percentage of positively stained cells. * *p* < 0.05, ** *p* < 0.01, *** *p* < 0.001. IgG served as a negative control.

**Figure 8 ijms-26-03785-f008:**
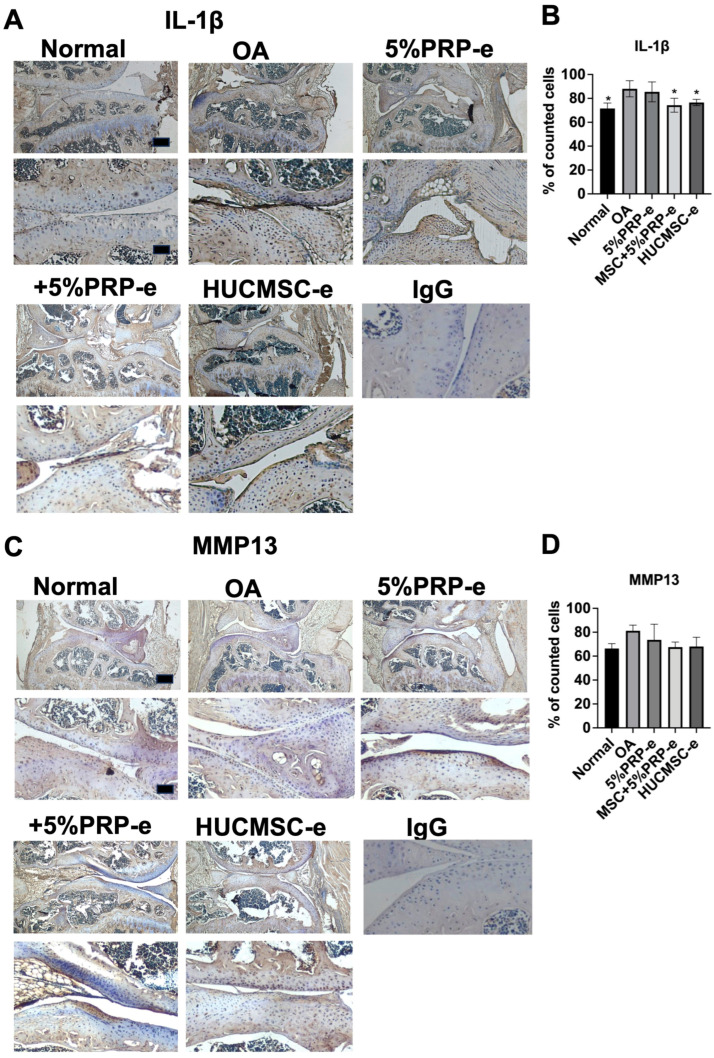
Immunohistochemistry of IL-1β and MMP13 in the mouse osteoarthritis model after 28 days of experiments. (**A**) A representative image of IL-1β is illustrated from the normal knee, osteoarthritis (OA), 5%PRP-e, MSC + 5%PRP-e, and MSC-e groups (n = 3 each). Scale bar = 100 μm. The lower panel is the magnification of the upper panel. Scale bar = 250 μm. (**B**) Quantitative analysis of percentage of positively stained cells. (**C**) A representative image of MMP13 is illustrated from the normal knee, OA, 5%PRP-e, MSC + 5%PRP-e, and MSC-e groups (n = 3). The lower panel is the magnification of the upper panel. (**D**) Quantitative analysis of the percentage of positively stained cells. * *p* < 0.05. IgG served as a negative control.

**Table 1 ijms-26-03785-t001:** The primers and product sizes of adipogenic, osteogenic, and chondrogenic genes by qRT-PCR analysis.

Gene	Forward	Reverse	Bp
*FABP4*	ATGGGATGGAAAATCAACCA	GTGGAAGTGACGCCTTTCAT	87
*PPARγ*	CCAGAAAGCGATTCCTTCAC	TGCAACCACTGGATCTGTTC	240
*RUNX2*	CGGAATGCCTCTGCTGTTAT	TTCCCGAGGTCCATCTACTG	174
*ALPL*	CCACGTCTTCACATTTGGTG	GCAGTGAAGGGCTTCTTGTC	96
*COL2A1*	GAGAGGTCTTCCTGGCAAAG	AAGTCCCTGGAAGCCAGAT	118
*ACAN*	GAGATGGAGGGTGAGGTC	ACGCTGCCTCGGGCTTC	443
*GAPDH*	GGTCTCCTCTGACTTGAACA	GTGAGGGTCTCTCTCTTCCT	221

## Data Availability

Owing to data protection regulations, it is not permissible to share the information contained in this article.

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
