# Peer review of "Enhancing the Therapeutic Potential of Human Umbilical Cord Mesenchymal Stem Cells for Osteoarthritis: The Role of Platelet-Rich Plasma and Extracellular Vesicles"

_ijms, 2025, doi:10.3390/ijms26083785_

Round 1

Reviewer 1 Report

Comments and Suggestions for Authors

Dear authors I have only one major comment which I think is important for your excellent work.

Author Response

Comment 1. Ιn figure 2 the differentiation is only performed at the gene level with real time PCR; I think it would be appropriate to check at the protein level as well. To analyze by western blot analysis and the expression of ALP, ACAN, PRAR and confirm also at protein level.

Response 1: We thank the reviewer for the valuable comment and agree with the concern raised. While Western blot data were not included, we have provided supporting evidence through immunohistochemical staining for type II collagen and aggrecan, which are key markers of cartilage matrix synthesis. Additionally, Safranin O staining was used to demonstrate glycosaminoglycan levels following chondrogenic differentiation. For adipogenic and osteogenic differentiation, we included Oil Red O and Alizarin Red staining, respectively. We believe this combination of staining methods provides complementary evidence that helps to support our findings in the absence of Western blot data.

Reviewer 2 Report

Comments and Suggestions for Authors

This work contained loads of hard work done by the authors. However, the authors should be careful of the description and interpretation of results so that readers can understand their findings in a more comfortable way. 

Comments on the Quality of English Language

The authors should clarify all the abbreviations used in the manuscript. Also, they should check all the grammatical mistakes.

Author Response

This work contained loads of hard work done by the authors. However, the authors should be careful of the description and interpretation of results so that readers can understand their findings in a more comfortable way.

Comment 1. Scale bars of all microscopic images should be checked and shown clearly. Images in Figure 7 and 8 contained no scale bar.

Response 1: We thank the reviewer’s comment. We have made the scale bar clear in all microscopic images and added the unit of the scale bar in the figure legends. We have also added the scale bar in Figures 7 and 8. 

Comment 2. The authors did not explain why 5% of PRP were used in the entire study. Will there be a concentration effect of PRP against OA?

Response 2: We thank the reviewer’s comment. We have tried 3% and 5% of PRP’s effects, and we found 5% PRP was helpful in OA treatment. 

Comment 3. A more detailed explanation of the settings of groups, namely, serum-free (SF) medium, 5% platelet-rich plasma (PRP), or exosome-depleted fetal bovine serum (+exo(-)FBS), should be present.

Response 3: We thank the reviewer’s comment. We have given a detailed explanation of the groups. (pages 15 and 17, methods sections 5.4, 5.5, and 5.11)

Comment 4. The sizes and concentrations of EVs obtained in different culturing conditions were different. How did the authors decouple size and concentration effects during the investigation of the treatment efficacy against OA?

Response 4: We appreciate the reviewer’s insightful observation regarding the differences in sizes and concentrations of EVs obtained under different culturing conditions. In our study, we primarily aimed to assess the overall therapeutic efficacy of EV preparations derived from various treatment conditions (+5% PRP-e, HUCMSC-e, etc.) rather than isolate the individual contributions of EV size or concentration. While we did observe variability in EV characteristics, the treatment dosages administered in vivo/in vitro were standardized based on EV concentration.

We acknowledge that the biological effects of EVs can be influenced by both particle size and concentration. However, decoupling these variables was beyond the scope of the current study. In future work, we aim to fractionate EVs by size (e.g., using gradient ultracentrifugation or size exclusion chromatography) and evaluate dose-dependent effects to more precisely dissect the roles of EV size and concentration on treatment efficacy.

We have added a brief explanation of this consideration to the Discussion section to clarify this point for readers. (page 13, lines 285-298)

The statements read as”

The biological effects of EVs can be influenced by both particle size and concentration. Exosomes are extracellular vesicles ranging from 30-150 nm in size that play a crucial role in intercellular communication [42,43]. Their biological effects are influenced by particle size, with smaller exosomes showing faster uptake by target cells and increased cellular motility [44]. Exosomes contain various biomolecules, including proteins, nucleic acids, and lipids, which can affect recipient cell function [42,43]. Their non-cytotoxic nature and ability to carry diverse cargo make them promising candidates for drug delivery and personalized medicine [45]. Studies have shown that EV concentration impacts gene expression and protein levels in target cells, with different effects observed at low versus high concentrations [46]. Sample concentration is critical for obtaining reproducible data in nanoparticle tracking analysis of EVs, with increased variability at higher concentrations due to particle interactions [47]. These findings highlight the importance of considering EV concentration and size distribution in both research and potential clinical applications.”

Comment 5. How did the authors treat OA mice with EVs?

Response 5: We thank the reviewer’s comment. We have added a section to describe the procedure of EV injections. (page 18, section 5.15)

The statements read as”

5.15. Extracellular Vesicle Injection

Seven days after collagenase injection, mice in the three treatment groups received EVs at a concentration of 1 × 10⁷ particles/mL, delivered in 50 μL of PBS. The injections were performed under anesthesia using ketamine (50 mg/kg) and xylazine (15 mg/kg). EVs were administered at sites located beneath the infrapatellar ligament in both hind knees. Following the injections, mice were allowed free movement and ad libitum access to food and water in their cages.”

Comment 6. In Figure 5, the staining images were too blurry and low in resolution. Why?

Response 6: We thank the reviewer’s comment. We have provided a more clear version of the figure (Figure 5). 

Comment 7. Regarding Figure 7 and 8, more interpretations of the staining results should be made in the Discussion. Differences in staining among different groups were somehow difficult to be noted.

Response 7: We thank the reviewer’s comment. We have added the interpretation of the staining results in the discussion section. (page 12, lines234-246)

The statements read as”

Figure 7 demonstrates that IHC staining for type II collagen and aggrecan—key markers of cartilage matrix integrity—was markedly reduced in the OA group, confirming matrix degradation typical of osteoarthritis [24]. Notably, treatment with +5% PRP-e or HUCMSC-e significantly restored the expression of both type II collagen and aggrecan, suggesting a regenerative effect that supports cartilage repair. These findings imply that +5%PRP-e and HUCMSC-e can mitigate cartilage breakdown and actively promote matrix synthesis. Figure 8 further complements these observations by showing reduced IL-1β expression in the treatment groups, highlighting an anti-inflammatory effect [25]. However, the lack of significant differences in MMP-13 staining across groups suggests that while +5%PRP-e and HUCMSC-e can reduce inflammatory cytokines and enhance matrix protein expression, they may have limited efficacy in directly suppressing matrix-degrading enzymes [26]. These results collectively underscore the dual function of these treatments in modulating inflammation and supporting matrix regeneration.”

Comments on the Quality of English Language

The authors should clarify all the abbreviations used in the manuscript. Also, they should check all the grammatical mistakes.

Response:  We thank the reviewer’s comment. We have checked all the grammatical mistakes. We have also clarify all the abbreviation used in the mansucript.

Reviewer 3 Report

Comments and Suggestions for Authors

General evaluation and characteristics of the reviewed scientific article:

The article presents an interesting preclinical study on the synergistic effect of PRP and EVs vesicles derived from human umbilical cord mesenchymal cells (HUCMSCs) in the treatment of an experimental mouse model of OA. The paper contains valuable experimental data and well-described methods; however, it also contains several important limitations that should be corrected before publication. Below are my detailed comments and observations on the paper.

Minor comments:

Expanding the introduction to provide a more in-depth discussion of osteoarthritis would significantly enhance this section by highlighting the importance and relevance of the condition. Osteoarthritis development is influenced by various factors, including occupational strain, participation in sports, musculoskeletal injuries, obesity, and sex. Presenting a detailed overview of these contributing elements, supported by relevant references, would establish a strong contextual foundation for the topic. The following sources are recommended for inclusion in this section:

https://doi.org/10.3390/healthcare12161648

DOI: 10.1056/NEJMcp1903768

To make the introduction clearer and more complete, it is suggested that the description regarding PRP be expanded. It is worth considering that platelet-rich plasma (PRP) is an autologous concentrate containing numerous growth factors (including TGF-β, PDGF, VEGF) that promote tissue regeneration, have anti-inflammatory effects and stimulate chondrocytes to synthesize extracellular matrix. In the context of osteoarthritis (OA), PRP is primarily used to treat mild to moderate lesions, but its clinical efficacy remains a matter of debate due to differences in preparation methods and formulation composition. The use of PRP as an additive for culturing MSCs may affect their biological activity, including the profile of secreted extracellular vesicles (EVs), providing a potential basis for synergistic therapeutic effects in OA. I recommend adding references: Short-term effects of arthroscopic microfracturation of knee chondral defects in osteoarthritis; Microfracture combined with platelet rich plasma for cartilage injury: A meta analysis ; Platelet-Rich Plasma Augmentation to Microfracture Provides a Limited Benefit for the Treatment of Cartilage Lesions: A Meta-analysis;

Despite the statement in the title and in the conclusions that PRP has a synergistic effect with EVs, the data presented in the paper indicate that the therapeutic effect of EVs + PRP is comparable to EVs derived from other culture conditions (+exo(-)FBS), which undermines the validity of the synergy thesis. In my opinion, the authors should consider performing additional statistical analyses with an assessment of the significance of the differences between the PRP-e and exo(-)FBS-e groups and introducing a clear definition of “synergistic effect” in the context of the results obtained.

Functional and histological evaluation was conducted only until the 28th day of the experiment. This is too short a time to reliably assess the durability of articular cartilage regeneration. The authors should emphasize in the discussion the need for long-term studies assessing the stability of therapeutic effects and the possible progression of degeneration over time. Please also describe the limitations in greater detail.

Although Western blot and NTA data have been presented, there is no accurate characterization of the miRNA or cytokine content of EVs, which limits the ability to understand the mechanisms of action. In future studies, it would be worthwhile to expand the analysis of the molecular content of EVs using, for example, proteomic analysis or small RNA sequencing.

The collagenase-induced osteoarthrosis model does not fully reflect the chronic, degenerative nature of OA in humans. It should be mentioned in the discussion that the model used has limited translatability and it is worth considering extending it to other models (e.g. DMM - destabilization of the medial meniscus).

There are repetitions in the text (e.g., repeatedly describing the expression of MSC markers and their behavior after PRP), which affects the clarity of the text. Language editing is recommended with the goal of eliminating repetitions and simplifying some passages, especially in the “Results” and “Discussion” sections.

Congratulations on an interesting study and I wish you continued success.

Author Response

General evaluation and characteristics of the reviewed scientific article:

Comment 1. The article presents an interesting preclinical study on the synergistic effect of PRP and EVs vesicles derived from human umbilical cord mesenchymal cells (HUCMSCs) in the treatment of an experimental mouse model of OA. The paper contains valuable experimental data and well-described methods; however, it also contains several important limitations that should be corrected before publication. Below are my detailed comments and observations on the paper.

Response 1: We thank the reviewer’s comment. We have revised the article with the comments. 

Minor comments:

Comment 2. Expanding the introduction to provide a more in-depth discussion of osteoarthritis would significantly enhance this section by highlighting the importance and relevance of the condition. Osteoarthritis development is influenced by various factors, including occupational strain, participation in sports, musculoskeletal injuries, obesity, and sex. Presenting a detailed overview of these contributing elements, supported by relevant references, would establish a strong contextual foundation for the topic. The following sources are recommended for inclusion in this section:

https://doi.org/10.3390/healthcare12161648

DOI: 10.1056/NEJMcp1903768

Response 2: We thank the reviewer’s comment. We have added a more in-depth introduction to osteoarthritis and two suggested references. (pages 1-2, lines 40-45)

The statements read as”

Although cartilage loss is central to the condition, OA affects the entire joint structure [3]. Knee OA presents symptoms such as pain, stiffness, limited joint mobility, and muscle weakness. Over time, it can lead to decreased physical activity, physical deconditioning, disrupted sleep, fatigue, depression, and increased disability [3]. Various factors, including occupational strain, participation in sports, musculoskeletal injuries, obesity, and sex, influence OA development [3,4].”

Comment 3. To make the introduction clearer and more complete, it is suggested that the description regarding PRP be expanded. It is worth considering that platelet-rich plasma (PRP) is an autologous concentrate containing numerous growth factors (including TGF-β, PDGF, VEGF) that promote tissue regeneration, have anti-inflammatory effects and stimulate chondrocytes to synthesize extracellular matrix. In the context of osteoarthritis (OA), PRP is primarily used to treat mild to moderate lesions, but its clinical efficacy remains a matter of debate due to differences in preparation methods and formulation composition. The use of PRP as an additive for culturing MSCs may affect their biological activity, including the profile of secreted extracellular vesicles (EVs), providing a potential basis for synergistic therapeutic effects in OA. I recommend adding references: Short-term effects of arthroscopic microfracturation of knee chondral defects in osteoarthritis; Microfracture combined with platelet rich plasma for cartilage injury: A meta analysis ; Platelet-Rich Plasma Augmentation to Microfracture Provides a Limited Benefit for the Treatment of Cartilage Lesions: A Meta-analysis;

Response 3: We thank the reviewer’s comment. We have added the reviewer’s comment to our introduction section. (page 2, lines 66-74)

The statements read as”

PRP is an autologous concentrate containing numerous growth factors (including transforming growth factor beta, platelet-derived growth factor, and vascular endothelial growth factor) that promote tissue regeneration, have anti-inflammatory effects, and stimulate chondrocytes to synthesize extracellular matrix [16,17]. PRP is primarily used in OA to treat mild to moderate lesions [17]. Still, its clinical efficacy remains debatable due to differences in preparation methods and formulation composition [17]. PRP used as an additive for culturing MSCs may affect their biological activity, including the profile of secreted extracellular vesicles (EVs), providing a potential basis for therapeutic effects in OA [17].”

Comment 4. Despite the statement in the title and in the conclusions that PRP has a synergistic effect with EVs, the data presented in the paper indicate that the therapeutic effect of EVs + PRP is comparable to EVs derived from other culture conditions (+exo(-)FBS), which undermines the validity of the synergy thesis. In my opinion, the authors should consider performing additional statistical analyses with an assessment of the significance of the differences between the PRP-e and exo(-)FBS-e groups and introducing a clear definition of “synergistic effect” in the context of the results obtained.

Response 4: We thank the reviewer’s comment. We deleted “synergistic” in the manuscript in response to the reviewer's comment and focused on the potential role of PRP in osteoarthritis treatment. 

Comment 5. Functional and histological evaluation was conducted only until the 28th day of the experiment. This is too short a time to reliably assess the durability of articular cartilage regeneration. The authors should emphasize in the discussion the need for long-term studies assessing the stability of therapeutic effects and the possible progression of degeneration over time. Please also describe the limitations in greater detail.

Response 5: We thank the reviewer’s comment. We have extended this part in the discussion section. (page 13, lines 299-309) 

The statements read as”

Recent studies have highlighted the need for long-term follow-up of MSC therapy for OA. A seven-year study on knee OA patients treated with adipose-derived MSCs showed significant clinical improvements for up to 60 months, with sustained cartilage structural benefits at 84 months [48]. However, an ovine model study found that while intra-articular MSC injections decelerated OA progression, no significant differences were observed between treated and control groups after 12 weeks [49]. The development of disease-modifying OA drugs remains a priority [50]. Safety concerns have been addressed in a 30-month follow-up study of 91 patients treated with autologous adipose-derived stem cells and PRP, which reported no neoplastic complications and only minor, self-limiting side effects [51]. These findings underscore the importance of continued research into the long-term efficacy and safety of stem cell or EV treatments for OA.”

Comment 6. Although Western blot and NTA data have been presented, there is no accurate characterization of the miRNA or cytokine content of EVs, which limits the ability to understand the mechanisms of action. In future studies, it would be worthwhile to expand the analysis of the molecular content of EVs using, for example, proteomic analysis or small RNA sequencing.

Response 6: We thank the reviewer’s comment. We will perform the suggested experiments in future projects. 

Comment 7. The collagenase-induced osteoarthrosis model does not fully reflect the chronic, degenerative nature of OA in humans. It should be mentioned in the discussion that the model used has limited translatability and it is worth considering extending it to other models (e.g. DMM - destabilization of the medial meniscus).

Response 7: We thank the reviewer’s comment. We have added this part in the discussion section. (page 13, lines 310-321)

The statements read as”

Recent studies have explored various OA models to understand the disease's pathogenesis better and evaluate potential treatments. The collagenase-induced OA model in rats and mice is used to study the progression of OA and potential therapies. This model mimics slow-progressing human OA, characterized by cartilage degeneration and synovial inflammation [52,53]. The destabilization of the medial meniscus (DMM) model has been widely used in mice, inducing structural changes in articular cartilage, subchondral bone, and menisci [54]. This model has been successfully applied in humanized mice, offering a promising tool for testing immune-interacting therapies [55]. While animal models remain valuable, engineered tissue models are emerging as potential alternatives for studying OA pathogenesis and evaluating treatments [56]. These diverse approaches contribute to a more comprehensive understanding of OA and may improve therapeutic strategies.”

Comment 8. There are repetitions in the text (e.g., repeatedly describing the expression of MSC markers and their behavior after PRP), which affects the clarity of the text. Language editing is recommended with the goal of eliminating repetitions and simplifying some passages, especially in the “Results” and “Discussion” sections.

Response 8: We thank the reviewer’s comment. We have removed repetitions in the text in the result and discussion sections to make it clear. 

Comment 9. Congratulations on an interesting study and I wish you continued success.

Response 9: We thank the reviewer’s comment.